# Landsat 9 Geometric Commissioning Calibration Updates and System Performance Assessment

**Michael J. Choate [1],\*, Rajagopalan Rengarajan [2]** 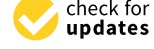**, James C. Storey [2] and Mark Lubke [2]**

[1] U.S. Geological Survey, Earth Resources Observation and Science Center, Sioux Falls, SD 57030, USA
[2] KBR, Contractor to the U.S. Geological Survey, Earth Resources Observation and Science Center, Sioux Falls, SD 57030, USA; rrengarajan@contractor.usgs.gov (R.R.); storey@contractor.usgs.gov (J.C.S.); mlubke@contractor.usgs.gov (M.L.)
\* Correspondence: choate@usgs.gov

**Abstract:** Starting with launch of Landsat 7 (L7) on 15 April 1999, the USGS Landsat Image Assessment System (IAS) has been performing calibration and characterization operations for over 20 years on the Landsat spacecrafts and their associated payloads. With the launch of Landsat 9 (L9) on 27 September 2021, that spacecraft and its payloads, the Operational Land Imager-2 (OLI-2) and Thermal Infrared Sensor-2 (TIRS-2), were added to the existing suite of missions supported by the IAS. This paper discusses the geometric characterizations, calibrations, and performance analyses conducted during the commissioning period of the L9 spacecraft and its instruments. During this time frame the following calibration refinements were performed; (1) alignment between the OLI-2 and TIRS-2 instruments and the spacecraft attitude control system, (2) within-instrument band alignment, (3) instrument-to-instrument alignment. These refinements, carried out during commissioning and discussed in this paper, were performed to provide an on-orbit update to the pre-launch calibration parameters that were determined through Ground System Element (GSE) testing and Thermal Vacuum Testing (TVAC) for the two instruments and the L9 spacecraft. The commissioning period calibration update captures the effects of launch shift and zero-G release, and typically represents the largest changes that are made to the on-orbit geometric calibration parameters during the mission. The geometric calibration parameter updates performed during commissioning were done prior to releasing any L9 products to the user community. This commissioning period also represents the time frame during which focus is more strictly placed on the spacecraft and instrument performance, ensuring that system and instrument requirements are met, as contrasted with the post commissioning time frame when a greater focus is placed on the products generated, their behavior and their impact on the user community. Along with the calibration updates discussed in this paper key geometric performance requirements with respect to geodetic accuracy, geometric accuracy, and swath width are presented, demonstrating that the geometric performance of the L9 spacecraft and its' instruments with respect to these key performance requirements are being met. Within the paper it will be shown that the absolute geodetic accuracy is met for OLI-2 and TIRS-2 with a margin of approximately 79% and 65% respectively while the geometric accuracy is met for OLI-2 and TIRS-2 with a margin of approximately 68% and 43% respectively.

**Keywords:** Landsat 9; Operational Land Imager-2; Thermal Infrared Sensor-2; calibration; instrument; spacecraft characterization

## 1. Introduction

The Landsat program has a long history of providing the science community with high quality radiometrically and geometrically corrected imagery. With the development and launch of Landsat 7 (L7) and its Enhanced Thematic Mapper Plus (ETM+) instrument the U.S. Geological Survey USGS created an Image Assessment System (IAS) to monitor,

characterize, and calibrate the Landsat spacecrafts and their instruments along with assessing the quality of the products generated from these missions [1–3]. A critical portion of this activity with respect to geometric calibration and characterization occurs during the commissioning period that immediately follows the on-orbit activation of the instruments and the reaching of the instrument's operational settings such that the instruments can start collecting imagery on a routine basis. It is during this time frame where the geometric calibration parameters defining the spacecraft's and instrument's pointing alignment knowledge, and the instruments focal plane alignment, produce the most change. This is expected as a result of the shaking imparted during launch, the transition from a 1-G to a 0-G environment, and the adjustment to the on-orbit thermal environment. Adjustments to the prelaunch parameters are needed before the distribution of the imagery to the user community, ensuring the best available quality of products, while also ensuring the spacecraft and instruments are meeting their pre-defined spacecraft, instrument, and mission level requirements or specifications.

### 1.1. Instrument Characteristics

The launch of Landsat 9 (L9) and its two instruments, the Operational Land Imager-2 (OLI-2) and the Thermal Infrared Sensor-2 (TIRS-2), on 27 September 2021, allows for the Landsat Mission objective to provide another data set of continuous high quality science data to the remote sensing user community [4,5]. The geometric aspects of the L9 commissioning work mimic those which were performed during the Landsat 8 (L8) commissioning period for its two instruments, the Operational Land Imager-1 (OLI-1) and the Thermal Infrared Sensor-1 (TIRS-1) along with the L8 spacecraft. The geometric algorithms and steps involved in performing calibration and characterization were identical between the L8 and L9 missions, the only differences being in the initial prelaunch values associated with the two instruments, and in the magnitude of the change between the prelaunch and postlaunch calibration parameters. This paper discusses both the OLI-2 and TIRS-2 geometric calibration changes determined during the L9 commissioning period. Unlike the L8 geometric commissioning papers, which were split between the two instruments and contained material on the pre-launch calibration activities, this paper combines the results from the two instruments and refers to the L8 commissioning papers with regard to the pre-launch activities that were performed [2,3].

Both the OLI-2 and TIRS-2 instruments are pushbroom sensors, each made up of a set of separate but overlapping linear arrays called Sensor Chip Assemblies (SCAs) [6,7]. The combined coverage of these SCAs determines the full nominal 185 km across track instrument swath. OLI-2 collects Visible and Near Infrared (VNIR) and Short-Wave Infrared (SWIR) wavelengths at a 30-m ground distance along with a panchromatic (pan) 15-m band. The TIRS-2 sensor collects two long wave thermal spectral channels centered at 10.9 μm (μm) and 12 μm at an approximately 100-m ground distance. These bands will have the following designations within the paper for OLI-2, bands 1–9, refer to coastal aerosol, blue, green, red, near infrared NIR, SWIR-1, SWIR-2, pan, and cirrus bands while for TIRS-2 bands 10 and 11 refer to the 10.9-μm and 12-μm wavelength bands. The staggering of the SCAs for each instrument in the along track direction, which allows for SCA-to-SCA overlap, means the bands will be imaging locations on the Earth at different times and sensor angles, requiring excellent spacecraft stability and ancillary data accuracy for good instrument and band registration. The differing angular viewing geometry for each SCA determines that terrain corrected imagery is needed to remove the parallax of each SCA in order to perform characterization and calibration while creating seamless data products [8]. Figures 1 and 2 shows the focal plane layout for the OLI-2 and TIRS-2 instruments. This staggering of the SCAs shown in Figures 1 and 2 also produces timing differences between when each band within a given SCA sees the same location on the ground. The time between when all the bands will image a given target on the Earth are 1.2 s for OLI-2 and 1.8 s for TIRS-2.

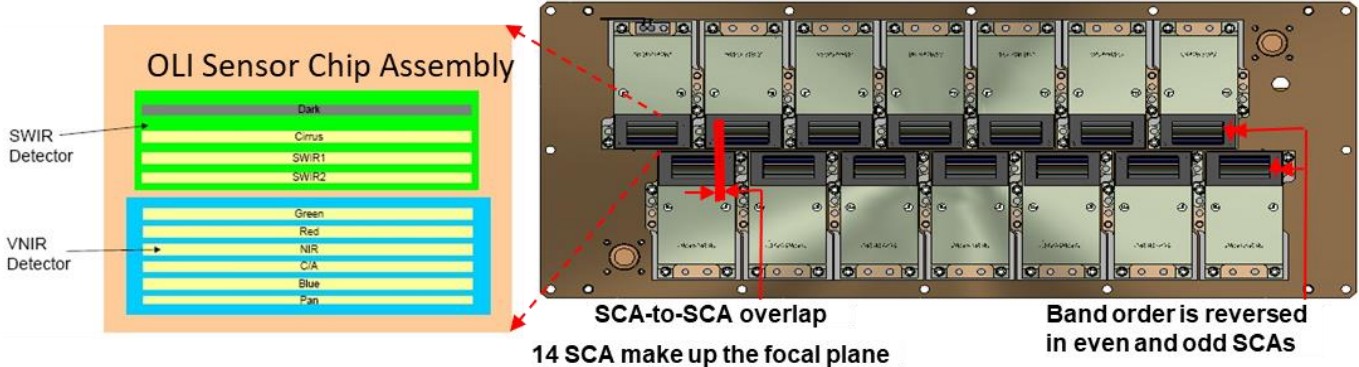

**Figure 1.** OLI-2 focal plane. Bands are staggered in the along track direction. The nominal 185 km field of view is achieved by 14 Sensor Chip Assemblies (SCAs) in the across track direction.

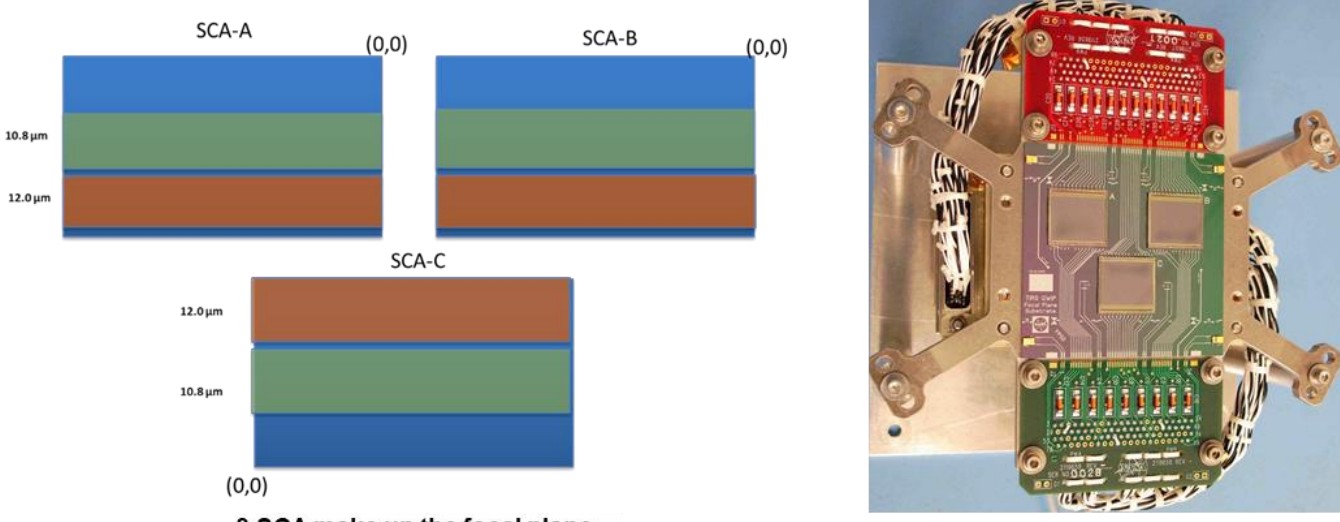

**Figure 2.** TIRS-2 Focal Plane. The two bands, a 10.9 μm and 12 μm band, are staggered in the along track direction. The nominal 185 km field of view is achieved by three Sensor Chip Assemblies (SCAs) in the across track direction.

### 1.2. Geometric Modelling Approach

A key aspect of the geometry of each instruments focal plane architecture is the modelling of their individual SCAs for each band. This modelling is done through a set of 3rd and 2nd order Legendre polynomials, where TIRS-2 uses a set of 3rd order polynomials and OLI-2 uses a set of 2nd order polynomials, which determine each detector's along-and across-track Line-of-Sight (LOS) vector components within an SCA for each band. These Legendre polynomials provide a convenient way to rapidly compute the LOS vector for any integer or non-integer detector location. Calibration of each instrument's focal plane involves determining an updated set of Legendre polynomial coefficients, with respect to pre-calibration values, based on measurements comparing established ground calibration sites to the Level-1 Precision Terrain Corrected Product (L1TP) imagery acquired over these sites. Updates to the Legendre polynomial coefficients are calculated to reduce the residual differences between the instruments' imagery and the calibration site reference imagery using a least squares approach, producing updated calibration parameters for use in product generation [9–11]. These Legendre coefficients are stored within a file, the Calibration Parameter File (CPF), which contains the mission's relevant geometric and radiometric calibration parameters. The CPF is maintained and updated by the IAS during normal operations. The CPF is a key input to the product generation process,

ensuring that geometrically calibrated and well registered products are distributed to the user community [12].

The Legendre equations operate on the detector number, normalized to span the range −1 to +1 over which the Legendre polynomials are orthogonal. These equations express the relationship between nominal (i.e., neglecting even-odd detector offset) normalized detector, for each band, for each SCA, and the X (along-track) and Y (across-track) components of the corresponding LOS vector. The form of the Legendre functions used to model LOS for the TIRS-2 and OLI-2 is show below. The additional third order term is used only in the modelling of the TIRS-2 LOS to account for the larger portion of the instrument field of view covered by each TIRS-2 SCA as compared to the OLI-2 SCA.

$$
\begin{aligned}
x = A_{0,m,n} + A_{1,m,n} * normalized\ detector + A_{2,m,n} * \left(\tfrac{3}{2} * normalized\ detector^{2} - \tfrac{1}{2}\right) + \\
A_{3,m,n} * \left(\tfrac{5}{2} normalized\ detector^{3} - \tfrac{3}{2} * normalized\ detector\right)
\end{aligned}
\tag{1}
$$

$$
\begin{aligned}
y = B_{0,m,n} + B_{1,m,n} * normalized\ detector + B_{2,m,n} * \left(\tfrac{3}{2} * normalized\ detector^{2} - \tfrac{1}{2}\right) + \\
B_{3,m,n} * \left(\tfrac{5}{2} normalized\ detector^{3} - \tfrac{3}{2} * normalized\ detector\right)
\end{aligned}
\tag{2}
$$

$$
normalized\ detector = \frac{2 * current\ detector}{number\ detectors - 1} - 1
$$

$$
A_{c,m,n} = Along\ Track\ Legendre\ for\ band\ number\ m,\ SCA\ number\ n
$$

$$
B_{c,m,n} = Across\ Track\ Legendre\ for\ band\ number\ m,\ SCA\ number\ n
$$

Equations (1) and (2). Legendre polynomials defining the Line of Sight (LOS) for the Operational Land Imagery, (OLI) 1 and 2, and the Thermal Infrared Sensors (TIRS), 1 and 2, aboard the Landsat 8 and Landsat 9 Satellites. Each set of polynomials defines a given detector, band, and Sensor Chip Assemblies (SCA) LOS vector.

Figures 3 and 4 show each detector's angular LOS location for the OLI-2 and TIRS-2 instruments, given in terms of the along and across track direction for each SCA of each band. The values shown in these figures represent the nominal LOS based on the post-commissioning updates performed on the pre-launch LOS values. For TIRS-2 its' SCAs are often referred to as A, B or C along with a numbering of 1–3. This is shown on Figure 4 with the A-B-C designation following a 3-1-2 numbering sequence.

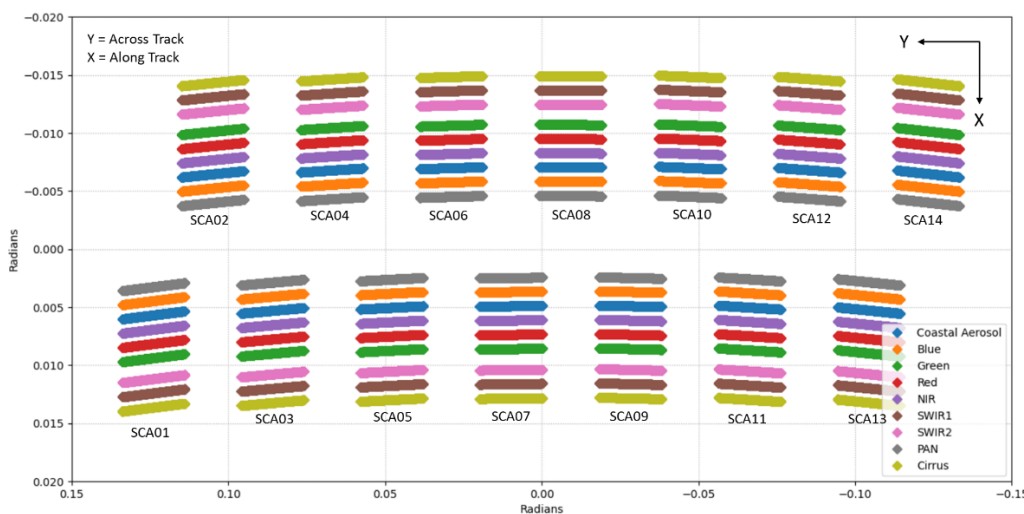

**Figure 3.** Operational Land Imager-2 (OLI-2) nominal detector Line-of-Sight (LOS) angular projection of the focal plane. The Sensor Chips Assembly (SCA) for each band is shown. Results shown are post-commissioning and represent the final calibration numbers at the end of the commissioning period.

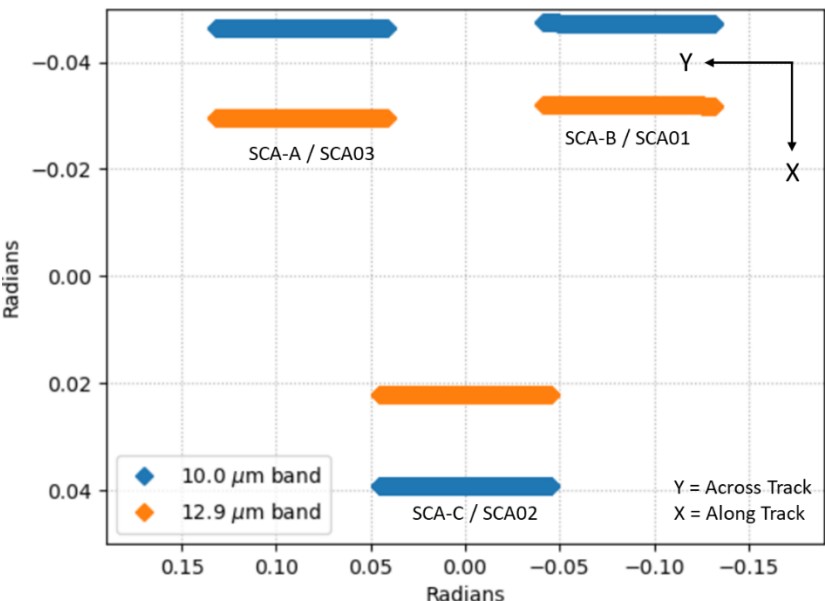

**Figure 4.** Thermal Infrared Sensor-2 (TIRS-2) nominal detector Line of Sight (LOS) angular projection of the focal plane. Each Sensor Chip Assembly (SCA) for each band is shown. Results shown are post-commissioning and represent the final calibration numbers at the end of the commissioning period.

The absolute accuracy of an L1TP product is dependent upon the accuracy of the ground control used to perform geometric correction, so this specification explicitly assumed that a set of ground control points accurate to 3 m 90% circular error (CE90) horizontally and 12 m 90% linear error (LE90) vertically were available for use in geometric correction, calibration, and accuracy verification [13,14]. One item that makes the commissioning activities challenging for geometric calibration is the fact that the satellite is not placed within its operational orbit following the Worldwide Reference System-2 (WRS-2) scene based global notation system, until several weeks after launch [15]. A carefully planned set of thruster burns are performed to raise the satellite's orbit into the final WRS-2 orbit grid system with the proper 8-day phasing relative to L8. This means that much of the early imagery acquired by the mission is not aligned with the WRS-2 reference grid. For the geometric calibration of the satellite and its instruments, imagery termed as Geometric Supersites that are derived from highly accurate high-resolution images, are used as a reference post-launch calibration source. These high resolution sets of imagery, mosaiced to provide more than a nominal WRS-2 geographic extent, allow mensuration of a given scene such that the full focal plane of the instrument can be assessed and characterized. This ability to measure the full field of view of the instrument over its operational environment can help alleviate limitations of performing instrument pre-launch ground calibration activities where a space environment can only be simulated and obtaining a full field of view of the instrument across multiple detectors can be difficult. To provide continuity across all Landsat missions these Geometric Supersites are used for all the Landsat instruments and spacecraft [16]. A limitation of these Geometric Supersites is that their locations are very sparse from a global perspective and typically only cover a little more than one WRS-2 path/row in geographic extent. This limitation means early image acquisitions, prior to achieving the final WRS-2 orbit, will frequently not have ground reference imagery that covers the full field of view of the instruments, limiting the ability to calibrate across the full focal plane of the instruments within any given single image acquisition. An example of an early in orbit image versus an image acquired on the WRS-2 grid is shown in Figure 5. The red outline shows a nominal WRS-2 image extent over the WRS-2 path/row. As the ground calibration sites typically cover only a slightly larger portion of the nominal WRS-2 geographic extent, extended only to account for orbital drift during normal operations, it can be seen from the figure that for the image acquisitions early in orbit the full field of

view will not be obtained for these acquisitions limiting the range of the focal plane that can be calibrated for this site.

**LC91030762021318LGN01**  **LC91030762021342LGN01**

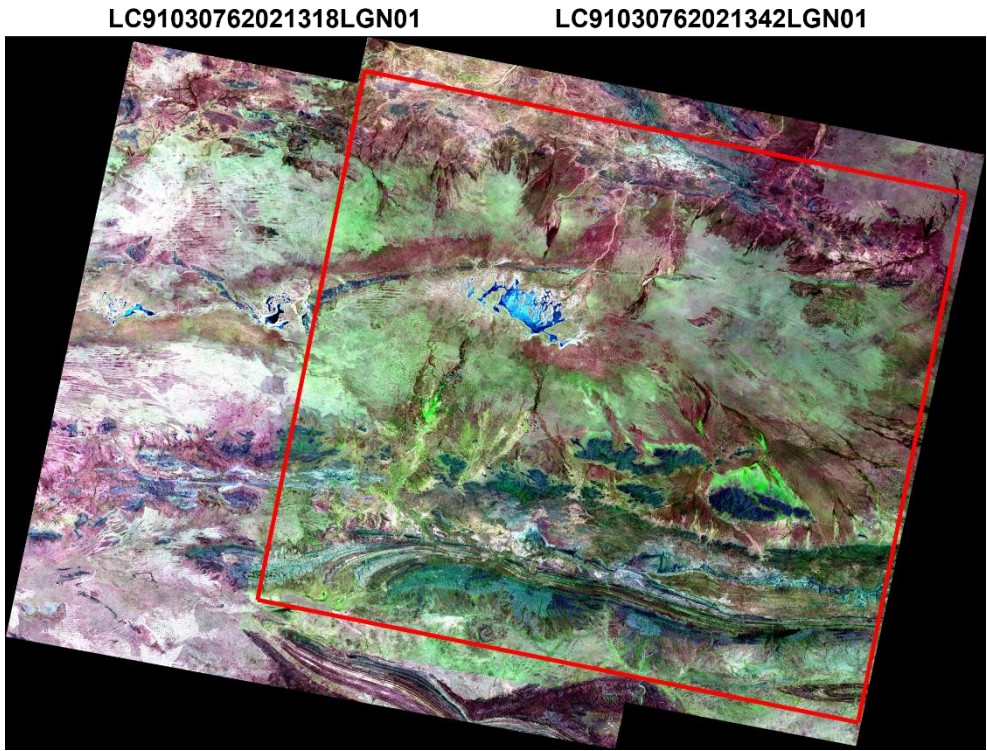

**Figure 5.** Landsat 9 pre World Reference System-2 (WRS-2) image acquisition and an acquisition once the spacecraft has reached its final WRS-2 orbit. The imagery marked with the scene identifier LC91030762021318LGN01 on the left was acquired prior to the satellite being set in its nominal WRS-2 orbit. The imagery marked with the scene identifier LC91030762021342LGN01 on the right was acquired with the satellite residing in its nominal WRS-2 orbit.

Although the pre WRS-2 orbit early in the mission produced challenges for obtaining L9 imagery that was completely contained within a given Geometric Supersite, the lower orbit itself was appropriately scaled during all calibration and characterization activities, where need, such that it did affect the outcome on any of the results produced during commissioning.

The location of the Geometric Supersites globally is shown in Figure 6. This imagery is composed of either USGS Digital Orthophoto Quadrangle (DOQ) data, from Global Positioning System (GPS) controlled Satellite pour l'Observation de la Terre (SPOT) imagery or from the Australian Geographic Reference Image (AGRI) provided by Geoscience Australia [17]. Although not used for calibration, the Landsat Global Land Survey (GLS) ground control is also used for a further verification of the calibration steps and shown in Figure 7, demonstrating both its distribution and the density. The Geometric Supersites are estimated to provide an absolute accuracy of approximately 6 m while the GLS ground control is estimated to provide an absolute accuracy of approximately 9 m [18–21].

As discussed previously, the pre WRS-2 acquisition grid during the beginning of the commissioning period can create a less than desirable situation of using WRS-2 centered reference images to calibrate only those portions of the full field of view of a given instrument contained within the reference image. This led to challenges in acquiring cloud-free calibration images with sufficient reference image overlap during the L8 commissioning work. This problem was mitigated for L9 by assembling multiple Geometric Supersites, including reference images, that were oversized in the east-west direction to provide complete coverage no matter where the spacecraft ground track fell. This made it easier to

collect the data necessary to both calibrate the sensors and to characterize the state of the spacecraft and instrument over time.

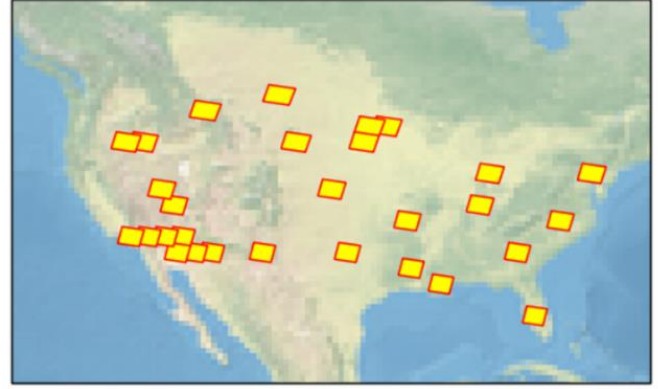

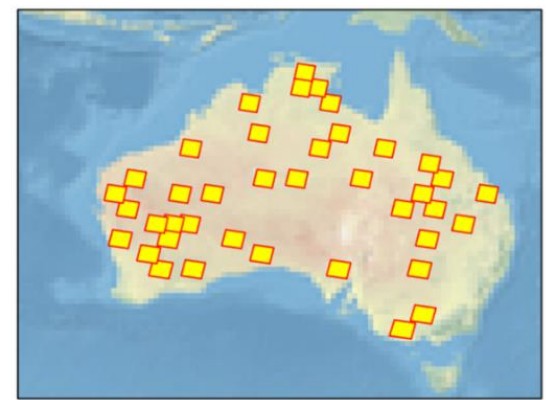

**Figure 6.** Distribution of Geometric Supersites. Images are mosaics of high-resolution Digital Ortho Photo Quadrangles, Satellite pour l'Observation de la Terre, or Geoscience Australia Reference Imagery. Images are of 15-m resolution from which 15 m Ground Control Chips are extracted.

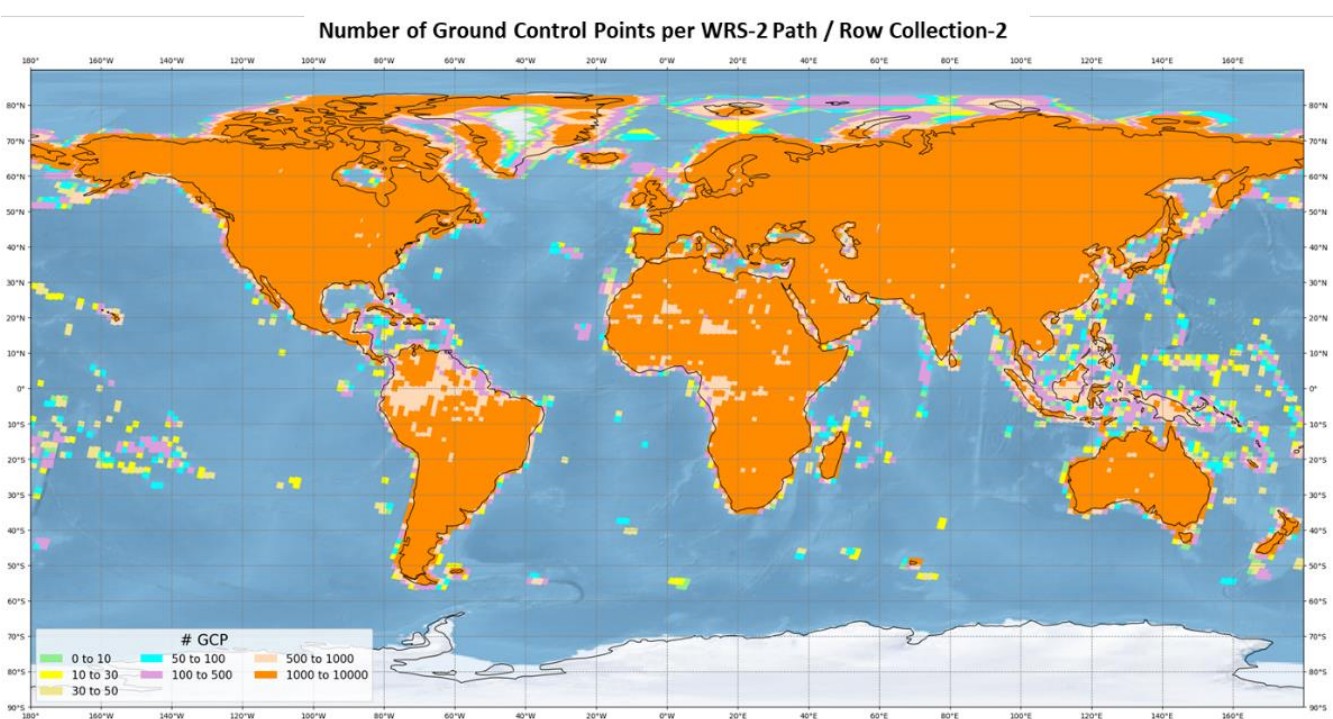

**Figure 7.** Density of the Global Land Survey (GLS) Ground Control Chips (GCPs). Chips are 30 m in resolution. These GCPs are used in generating USGS Landsat products for distribution to the user community.

## 2. Materials and Methods

Prior to the launch of L7 and its ETM+ sensor on 15 April 1999, the USGS developed and built the IAS to calibrate and characterize the L7 spacecraft and the ETM+. Since that time, the mission of the IAS has expanded to include all Landsat missions and instruments with respect to calibration and characterization. This expansion included the ability to calibrate and characterize the L8 spacecraft and its instruments the OLI-1 and the TIRS-1, and then in late 2021 the L9 and the OLI-2 and TIRS-2 instruments. Due to the similarity

in the spacecrafts and instruments the steps involved, the L9 commissioning and its instruments were the same as that of L8. This paper briefly describes the steps involved with the commissioning of L9 from a geometry perspective. Further information on the calibration steps performed during commissioning can be found in the L8 and L9 Algorithm Description Documents (ADD) [22].

The geometric calibration of the L9 spacecraft and its instruments involve a specific set of steps that need to be performed in a specific order. Because many of the geometric calibration procedures rely on being able to register new OLI-2 acquisitions to ground reference images, an initial coarse alignment calibration between the OLI-2 pan band and the spacecraft attitude reference is used to achieve an absolute instrument pointing accuracy sufficient to allow automated ground control point mensuration. The basis for the absolute calibration of all instruments onboard L9 relies on a calibrated OLI-2 pan band, therefore the first priority for geometric calibration is to perform the OLI-2 Focal Plane Alignment on the pan band. With the pan band representing the highest resolution band among all instruments aboard L9, this band allows the best mensuration accuracy, and establishes a reference for all lower resolution bands. Once the OLI-2 pan band has its SCAs calibrated, both relative to each other and with respect to the attitude control reference system the next step is to align the OLI-2 multispectral bands to the OLI-2 pan band using the OLI-2 band-to-band alignment calibration procedure. Once the OLI-2 bands are calibrated, the next step is to improve the geometry of the TIRS-2 10.9 μm band by measuring it against the OLI-2 SWIR-1 band using the TIRS alignment calibration procedure. The OLI-2 SWIR-1 band is used as a reference for the TIRS instrument as this band's center wavelength, 1.6 μm, closely aligns with that of the TIRS-2 10.9 μm center wavelength. Although these two bands center wavelength, and band passes, are not an ideal match when performing the image correlation during the mensuration step, they have been found to produce the best results among all the band combinations. This step simultaneously improves the 10.9-μm band internal geometry by aligning the SCAs and registers the TIRS-2 band to the OLI-2 SWIR-1 band. Once the OLI-2 pan band is calibrated both absolutely and internally, the OLI-2 multispectral bands are aligned to the pan band, and the TIRS-2 10.9-μm band is aligned to OLI-2 instrument, the final geometric calibration step is for the TIRS-2 10.9-μm and 12-μm bands to be aligned (TIRS-2 band-to-band alignment). A step that was mentioned briefly above but is a key part of the initial and final Focal Plane Alignment, involves aligning the pan band's optical axis to the spacecraft's Attitude Control System (ACS). This OLI-2 focal plane to ACS alignment, which calibrates the overall absolute pointing of the pan band, allows in turn, an absolute pointing calibration to be transferred to all the other lower resolution bands (including the TIRS-2 bands).

All analysis that follows within this paper is based on a set of acquisition dates from 31 October 2021 to 31 December 2021 which defines the geometric commissioning period for the L9 spacecraft and its instruments. The results shown that were measured with respect to the Geometric Supersites ground control, using the Geodetic and Geometric Accuracy assessments, are performed to assess whether the spacecraft, the instrument, and the system requirements are met. The results shown involving the GLS ground control, and the Geodetic and Geometric Accuracy assessments, are performed to demonstrate what can be expected from a geometric product accuracy standpoint. The Focal Plane Calibration procedure is based solely on Geometric Supersite locations whereas the band-to-band and instrument-to-instrument calibrations are based on both GLS and Geometric Supersite ground control. In these operations, performed only after Focal Plane Calibration has been completed, the slightly degraded accuracy when using the GLS ground control does not affect results. All calibration and characterization steps are performed based on a scene selection that is dependent on being able to correlate and achieve good mensuration results across all band wavelengths used within the calibration steps and across the full extent of the imagery. This requires cloud-free imagery, typically less than 5%, and ground features that correlate well between the Geometric Supersites and OLI-2 pan band or that correlate

well across the corresponding band combinations that are being calibrated, for example, OLI-2 band 6 and TIRS-2 band 10 for OLI-to-TIRS alignment.

## 3. Results

The OLI-2 and TIRS-2 swath width have a requirement to cover a minimum of 185 km and for the TIRS-2 instrument to be contained within the field of view of the OLI-2 instrument. Related to the swath width, is a requirement for there not to be any data gaps between the adjacent SCAs within the bands of each instrument. The Landsat standard data acquisition schedule is dependent upon the sun elevation angle, eliminating high latitude scenes from being acquired during the commissioning time frame, which occurred over the winter months. Because the swath width and SCA-to-SCA overlap effects are both dependent upon within orbit location, as both are influenced by the satellite flight path angle and orbit height, the lack of high latitude scenes limited the range of available test data. As a result, scenes at mid-to-high latitudes and at the equator were inspected for swath width, between-sensor coverage, and SCA overlap. An example of one of the images inspected for these requirements is shown in Figure 8 where a combination of blue, SWIR-2, and the TIRS-2 10.9 μm band are shown as a red-green-blue (RGB) image. The overlap in the multispectral OLI-2 bands with respect to the TIRS-2 band can be seen in the figure, demonstrating that TIRS-2 is contained within the field of view of OLI-2. The measured swath width upon inspection of the imagery was found to be, at the equator, 189.96 km for OLI-2 and 186.66 km for TIRS-2. A caveat to this method in measuring an L1TP product for the instrument's swath width is that the L1TP imagery is trimmed by a few pixels at the extremes of the eastern and western portions of the image, when a data set is framed to a map projection and resampled to a nominal 30 m pixel spacing, the actual instrument swath width is thus slightly larger than this visually measured coverage extent. For determining the overlap between adjacent SCAs, the Legendre LOS polynomials were first inspected for the angular overlap between adjacent SCAs, a plot of which is shown in Figure 9 for the OLI-2 instrument. This analysis found at least 23 multispectral pixels of overlap between adjacent SCAs. The odd-to-even pattern in the plot is due to a small yaw component in the instrument alignment to the optical axis. As the between-SCA overlap is affected by the flight dynamics of the spacecraft due to altitude changes and a within orbit, latitude based, yaw steering is applied to the spacecraft, with the yaw steering providing a better alignment of the spacecraft's ground velocity vector to its direction of flight. A visual inspection was also performed to ensure adequate SCA-to-SCA overlap by looking for fill data between adjacent SCA boundaries. Through both a visual inspection and an inspection of the Legendre LOS coefficients, no gaps between SCAs within the individual bands were found. The same overlap analysis between adjacent SCAs was performed for the TIRS-2 instrument and those results are shown in Figure 10.

Absolute geodetic accuracy refers to the geolocation accuracy of geometrically corrected products prior to the application of ground control points, taking into account terrain effects, and is most often associated with the absolute pointing accuracy of the spacecraft and instrument. The absolute geodetic accuracy is primarily dependent on the accuracy of the spacecraft telemetry, specifically the spacecraft's ephemeris and attitude information, and the ability to align the OLI-2 focal plane optical axis to that of the spacecraft's ACS. For example, the parameters associated with alignment between the OLI-2 focal plane optical axis and the spacecraft ACS are determined through the Sensor Alignment algorithm. Relative geodetic accuracy refers to the internal accuracy of the geometrically corrected products prior to the application of ground control points while taking into account terrain effects. Both geodetic accuracy requirements are primarily dependent upon the spacecraft performance of its telemetry but also involve the OLI-2 instrument pointing knowledge (thermal) stability, which translates to its LOS stability over time, the ability to align the OLI-2 pan band to the spacecraft ACS, and the ability to align each SCA within each instruments focal plane. Once an SCA alignment for the pan band is performed, an OLI-2 instrument band-to-band alignment can be performed aligning all other OLI-2 bands and

their SCAs to the calibrated pan band. Once an alignment of the OLI-2 bands is achieved, the SCA alignment of the TIRS-2 thermal bands can be performed. The combination of these steps, specifically with the OLI-2 pan band being fully calibrated, achieves the final post-calibration geodetic accuracy results for the system of both instruments for all bands.

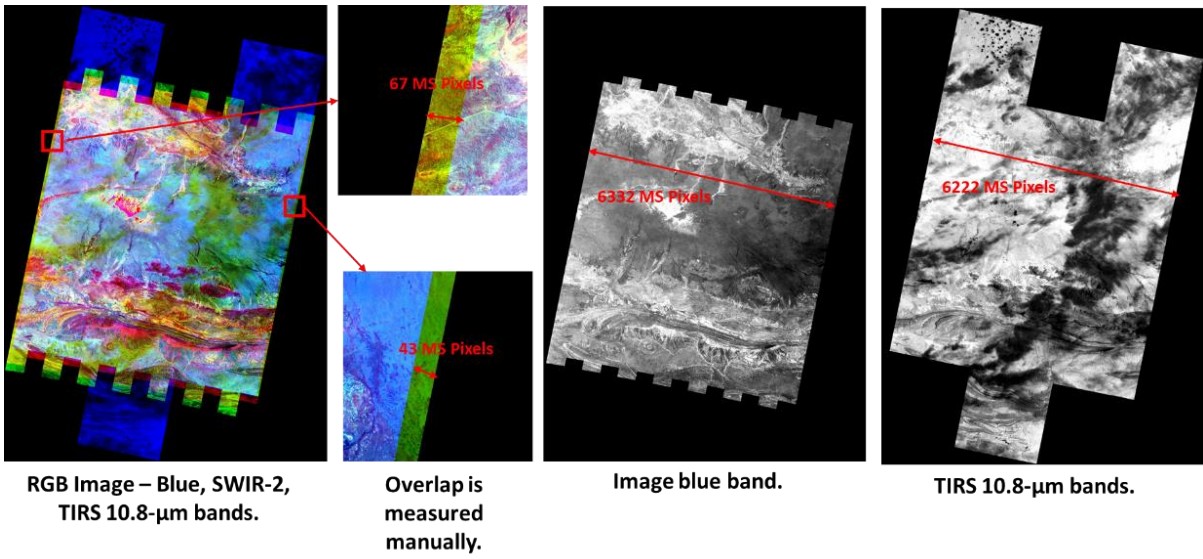

**RGB Image – Blue, SWIR-2, TIRS 10.8-µm bands.** **Overlap is measured manually.** **Image blue band.** **TIRS 10.8-µm bands.**

**Figure 8.** Red, green, blue (RGB) image created from the Operational Land Imager-2 (OLI-2) blue, Short Wave Infrared (SWIR-2) band and the Thermal Infrared Sensor-2 (TIRS-2) 10.9-µm band. This RGB image demonstrates that the TIRS-2 field of view is contained within that of the OLI-2 instrument.

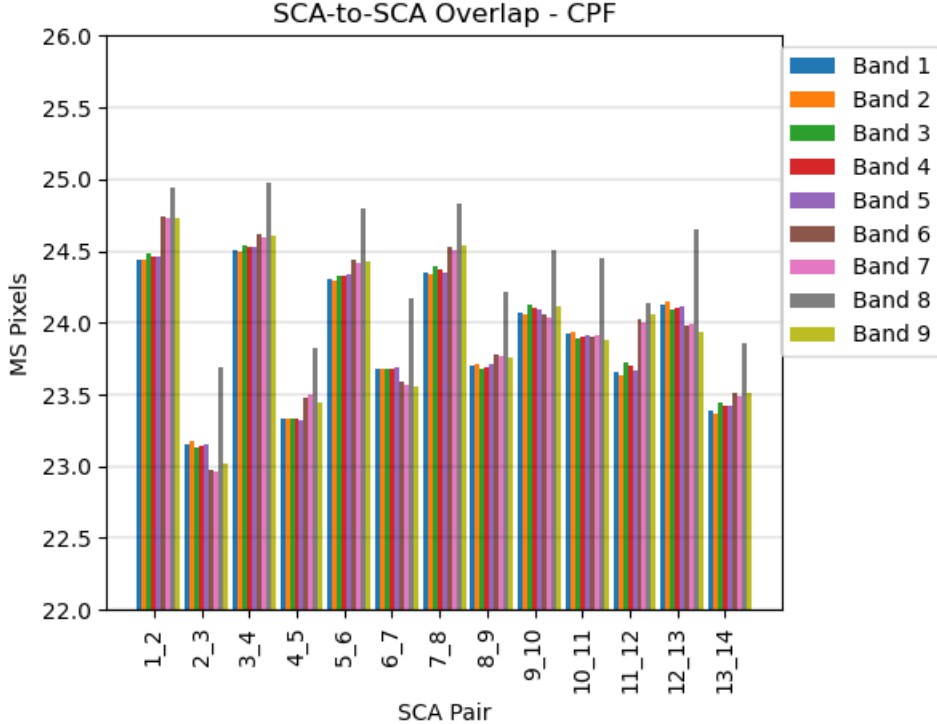

**Figure 9.** Operational Land Imager-2(OLI-2) Sensor Chip Assembly (SCA) overlap based on the Legendre Line of Sight (LOS) polynomials as listed in the Calibration Parameter File (CPF). The nominal outmost and innermost detectors for adjacent SCAs are compared to determine the amount of angular overlap between the two detectors.

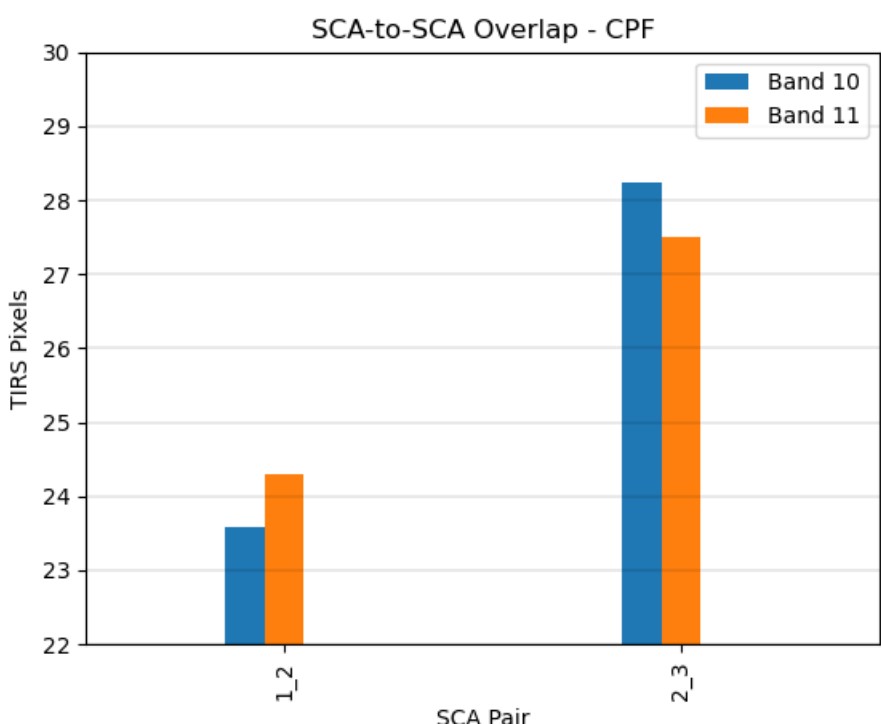

**Figure 10.** Thermal Infrared Sensor-2 (TIRS-2) Sensor Chip Assembly (SCA) overlap based on the Legendre Line of Sight (LOS) polynomials as listed in the Calibration Parameter File (CPF). The nominal outmost and innermost detectors for adjacent SCAs are compared to determine the amount of angular overlap between the two detectors.

The results from the Sensor Alignment calibration procedure, and the corresponding calculated roll, pitch, and yaw interior orientation parameters, determined both during pre-launch testing and during the commissioning period are shown in Table 1. With respect to the sensor alignment L9 requirements, the alignment control requirement is 4 milli-radian (mrad), the post-launch requirement for co-alignment between instruments is 7 mrad, and the alignment knowledge requirement is 2 mrad (pre-launch). All of these alignment requirements, pre-launch, and post-launch, were met (Table 1).

**Table 1.** Landsat 9 instrument roll, pitch, and yaw calibration alignment parameters in milli-radians determined pre-launch and post-launch. These values allow for the alignment between the spacecrafts Attitude Control System (ACS) and the Operational Land Imager-2 (OLI-2) instruments optical axis, giving a more accurate absolute pointing knowledge for the instrument.

| | Pre-Launch | | | Post Launch | | | Deltas | | |
|---|---|---|---|---|---|---|---|---|---|
| **Milli-Radians** | **Roll** | **Pitch** | **Yaw** | **Roll** | **Pitch** | **Yaw** | **Roll** | **Pitch** | **Yaw** |
| ACS to OLI | −0.0890 | 0.7682 | 0.7003 | 0.1856 | 1.1008 | 0.9266 | −0.2747 | −0.3326 | −0.2263 |
| ACS to TIRS | −1.4076 | 0.8644 | 1.8104 | −2.2011 | 0.7403 | 2.1731 | 0.7936 | 0.1242 | −0.3627 |
| TIRS to OLI | 1.3195 | −0.0946 | −1.1103 | 2.3877 | 0.3633 | −1.2457 | −1.0682 | −0.4579 | 0.1354 |
| | Alignment Requirements | | | | | | | | |
| Alignment Control | <4 | | | milli-radians | | | | | |
| Co-Alignment | <7 | | | milli-radians | | | | | |
| Alignment Knowledge | <2 | | | milli-radians | | | | | |

As discussed previously, Focal Plane Alignment aligns the pan band SCAs, from both an absolute and relative perspective, based on measurements between the L1TP imagery and Geometric Supersites. Once this step is performed an OLI-2 band-to-band calibration

can be performed, adjusting the OLI-2 multispectral bands to align those bands to the calibrated pan band, SCA-by-SCA. The combination of these two steps determines the new, post-launch, LOS calibration parameters for all the OLI-2 bands and their SCAs. This set of LOS calibration parameters are updates to the existing pre-launch Legendre Coefficients stored in the CPF. The mean adjustment for each of the OLI-2 bands LOS, in both the along and across track direction, between the pre-launch and the post-launch commissioning calibration parameters, are shown in Figures 11 and 12. The results shown are listed in micro (μ) radians where 42.5 μ radians is one nominal multispectral Instantaneous Field of View (IFOV). Figures 13 and 14 show the same mean adjustments but for each of the TIRS-2 bands. The nominal IFOV for the TIRS-2 bands is 141.86 μ radians.

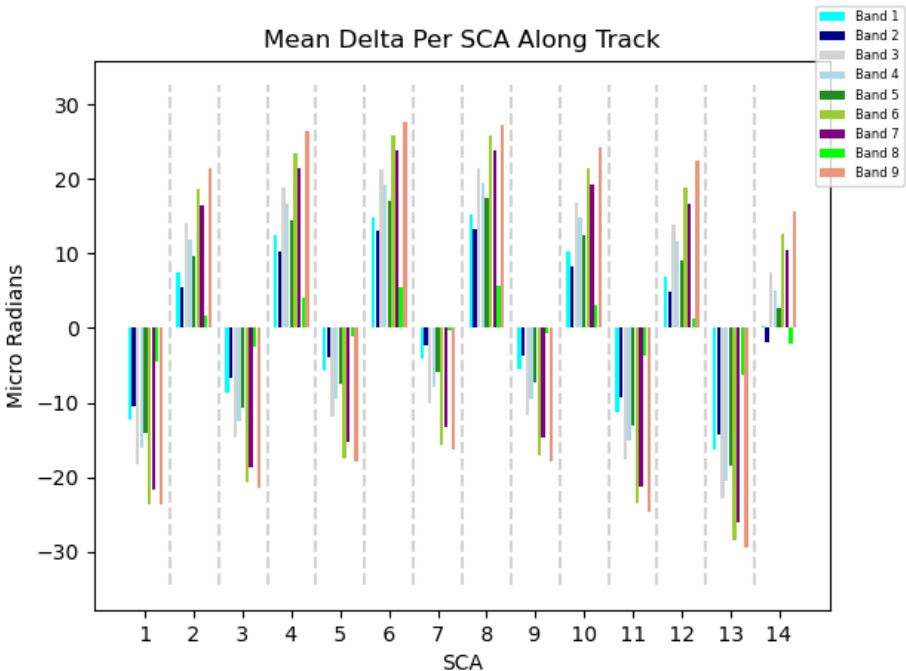

**Figure 11.** Mean deltas between pre-launch and post-commissioning along track Line of Sight (LOS) polynomials for each Sensor Chip Assembly (SCA) of each band of the Landsat 9 Operational Land Imager-2 (OLI-2). Values are given in terms of micro-radians. Adjustments post-commissioning were less than one multispectral OLI-2 pixel which has a nominal Instantaneous Field of View (IFOV) of 42.5 micro radians.

Focal Plane Calibration must be performed to ensure a fully calibrated pan band, which is needed to achieve calibration of the multispectral OLI-2 bands and the TIRS-2 bands. One method for quantifying the results of the Focal Plane Calibration, to ensure that adequate results have been obtained so that the other instrument bands can be calibrated, is to perform (in some cases recalculate) Focal Plane Calibration on the Geometric Supersites once the pan band has been fully calibrated. These results will give the post-fit residuals for the focal plane alignment procedure and are shown in Figure 15. The mean pre-fit (which would represent post-calibration) residuals of all the datasets is shown in the figure with error bars that represent the standard deviation of those means. From Figure 15 the pan band SCAs are fitted to the Geometric Supersite control with a mean error that is less than 2 μ radians and with a standard deviation of less than 6 μ radians. Considering the pan band has a nominal IFOV of 21.25 μ radians, and a nominal ground instantaneous field of view (GIFOV) of 15 m, the mean results show a fit of less than a tenth of a pixel to the reference imagery.

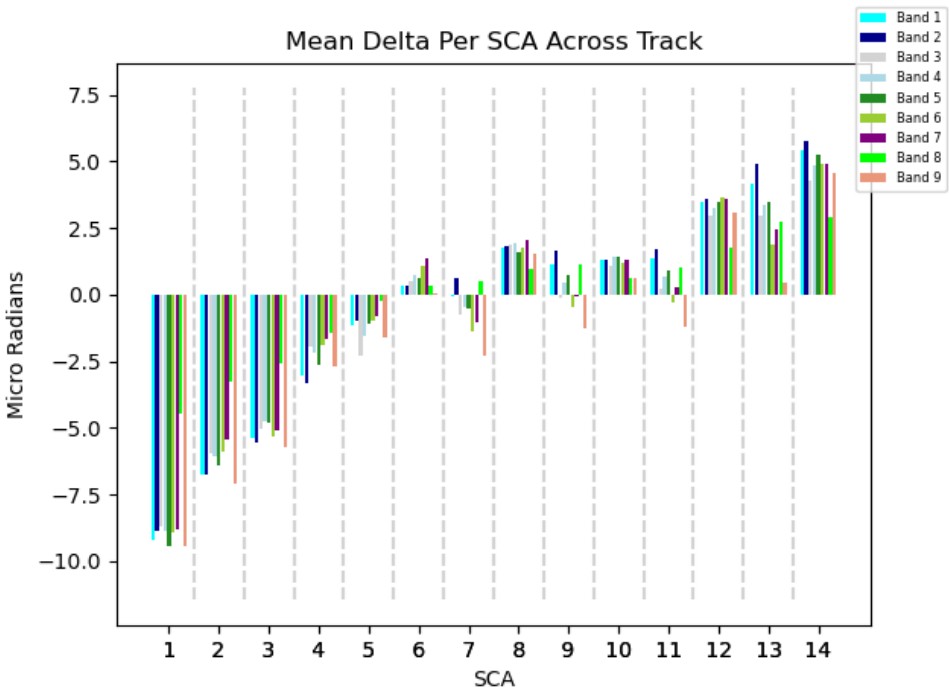

**Figure 12.** Mean deltas between pre-launch and post-launch across track Line of Sight (LOS) polynomials for each Sensor Chip Assembly (SCA) of each band of the Landsat 9 Operational Land Imager-2 (OLI-2). Values are given in terms of micro radians. Adjustments post-launch were less than one multispectral OLI-2 pixel which has a nominal Instantaneous Field of View (IFOV) of 42.5 micro radians.

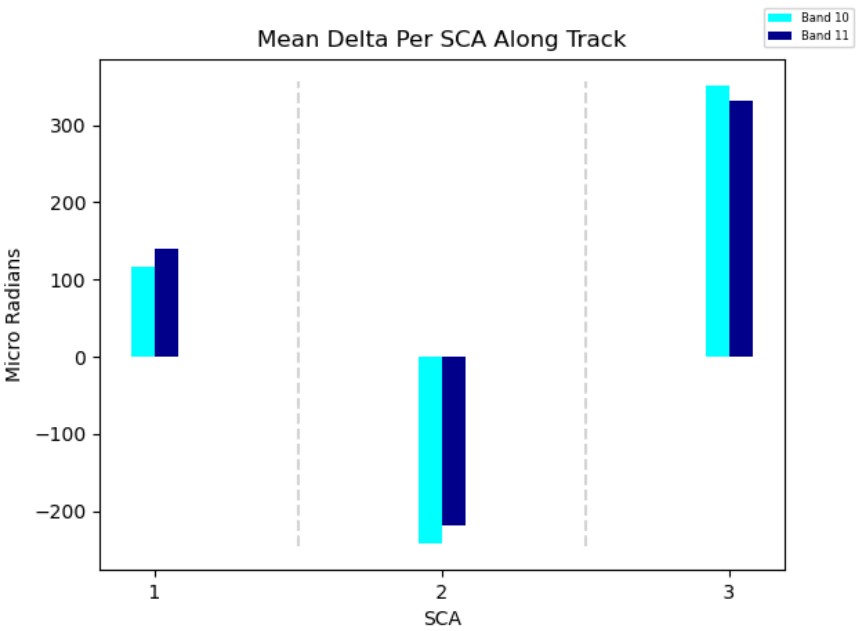

**Figure 13.** Mean deltas between pre-launch and post-launch along track Line-of-Sight (LOS) polynomials for each Sensor Chip Assembly (SCA) of each band of the Landsat 9 Thermal Infrared Sensor-2 (TIRS-2). Values are given in terms of micro radians. Adjustments post-launch were less than two thermal TIRS-2 pixels which has a nominal Instantaneous Field of View (IFOV) of 141.86 micro radians.

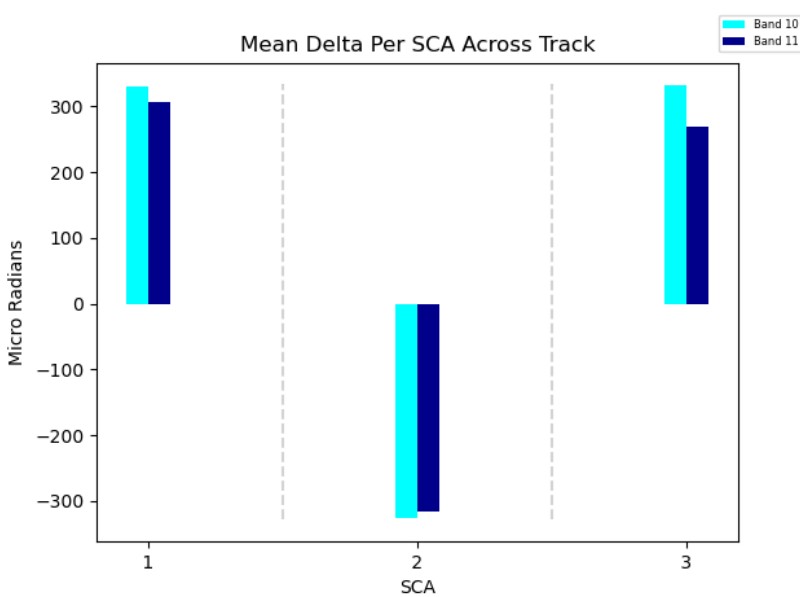

**Figure 14.** Mean deltas between pre-launch and post-launch across track Line-of-Sight (LOS) polynomials for each Sensor Chip Assembly (SCA) of each band for the Landsat 9 Thermal Infrared Sensor-2 (TIRS-2). Values are given in terms of micro radians. Adjustments post-launch were less than three thermal TIRS-2 pixels which has a nominal Instantaneous Field of View (IFOV) of 141.86 micro radians.

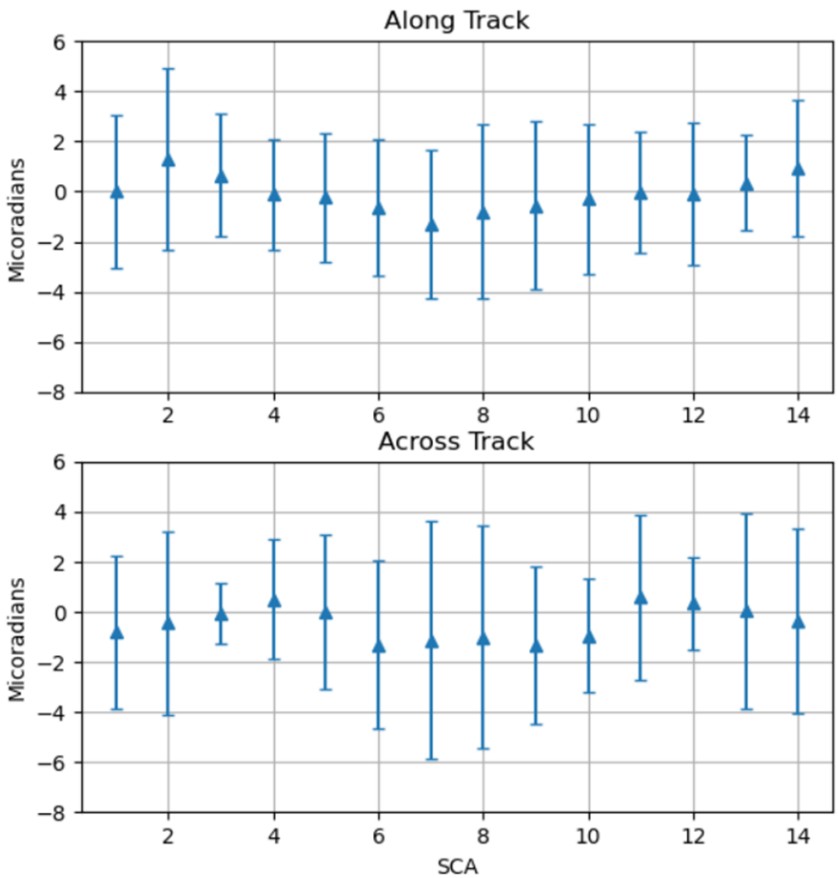

**Figure 15.** Focal Plane Calibration post-calibration mean residuals. The standard deviation of those means are shown as error bars. Results are based on running the calibrated data through the Focal Plane Alignment algorithm, with the remaining fit offsets demonstrating any residual alignment post-calibration.

A similar assessment can be done on the alignment of the TIRS-2 focal plane by running the TIRS-2 to OLI-2 alignment after calibration has been performed on both instruments, inspecting the pre-fit results generated. These results, representing an assessment of post-fit calibration of the TIRS-2 10 µ meter band SCAs, is shown in Figure 16.

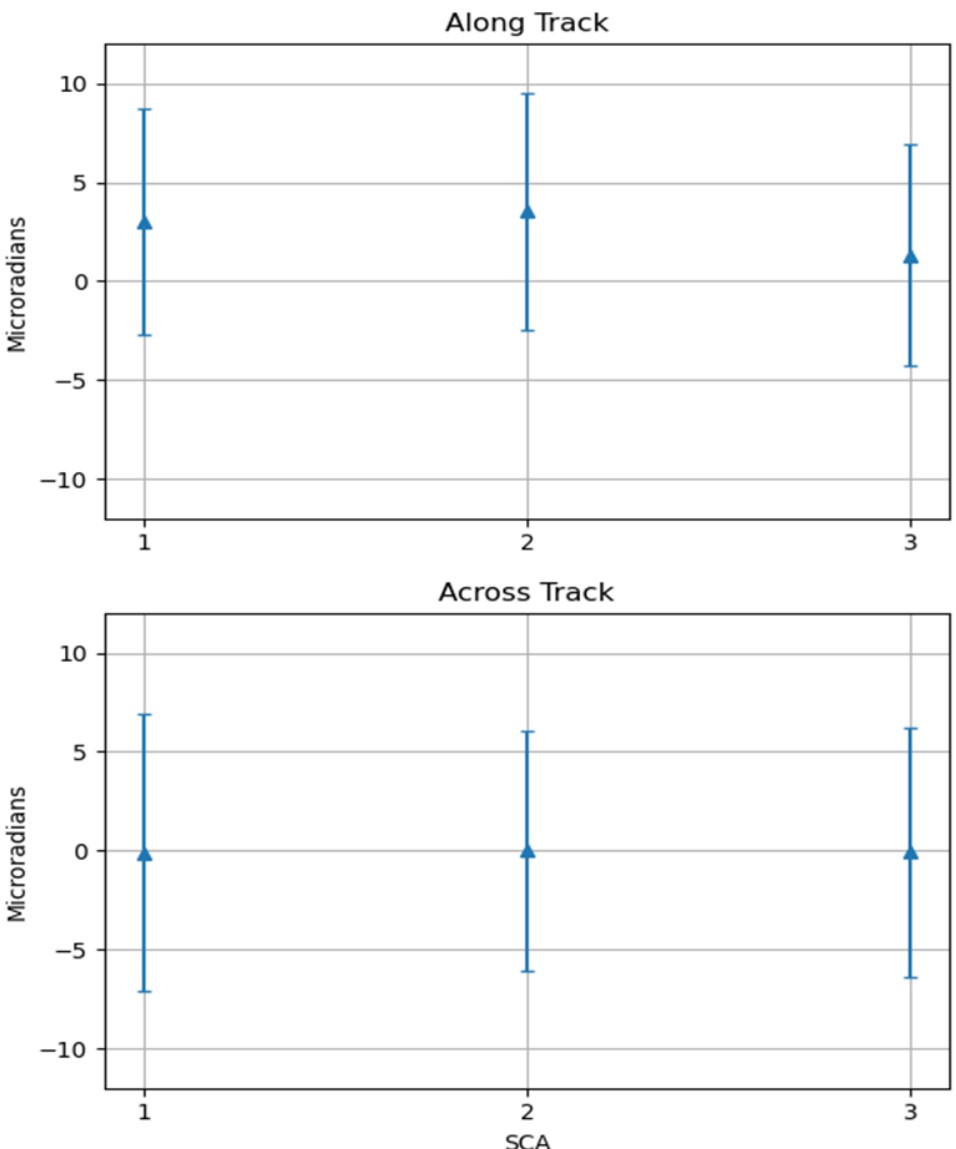

**Figure 16.** TIRS-2 to OLI-2 alignment post-calibration mean residuals. The standard deviation of the means are shown as error bars. Results are based on running the calibrated data through the TIRS-2 to OLI-2 Alignment algorithm, with the remaining fit offsets demonstrating any residual alignment post-calibration.

The mean values shown in Figures 15 and 16 for each SCA measured demonstrates the lack of any mean residual bias being present within the alignments performed for the two instruments. The standard deviations of those calculated means shown in both Figures 15 and 16, where a maximum value of approximately 9 µ radians for OLI-2 and 15 urad for TIRS-2 are shown, are within a range of what can be expected when using image correlation to measure the displacement between spatial data sets [23,24]. Comparing these standard deviations against the OLI-2 and TIRS-2 nominal IFOVs of 42.5 µ radians

and 141 μ radians also demonstrates that the magnitude of these measurements are small compared to the nominal pixel size of the instruments.

Geodetic accuracy pertains to two aspects of the spacecraft and instrument, the absolute pointing accuracy of the instrument and the relative internal, short-term, LOS stability of the instrument. Both sets of geodetic accuracy assessments apply to conditions prior to correcting the data for any position and attitude telemetry errors but involve removing terrain induced parallax offset within the imagery prior to the assessment. Absolute geodetic accuracy refers to the pointing knowledge of the spacecraft and instruments boresight, without the use of any external influences such as ground control. Relative Geodetic Accuracy refers to the internal accuracy of the imagery, the point-to-point relative accuracy within the scene. These geodetic accuracy assessments for OLI-2 are measured directly from the precision correction solution step, which calculates pre-fit statistics on measured ground control points, during the process of creating a precision-terrain corrected set of imagery from a select set of cloud-free scenes. For commissioning these assessments were performed using both the Geometric Supersite and the GLS ground control. The geodetic assessment using the Geometric Supersite ground control is carefully chosen to avoid clouds or any snow that may be present in the imagery that would degrade the measurements taken, this was accomplished by choosing scenes with less than 4% cloud cover, less than 4% snow cover and at times with a visual inspection of the data. For commissioning, 62 Geometric Supersites were chosen for assessment. The use of the more accurate Geometric Supersites and careful selection of cloud and snow free imagery for this assessment, allows for a more direct understanding of the spacecraft and instrument performance. The geodetic assessment using the GLS ground control includes a wider and more dense set of imagery, globally acquired, over the commissioning time frame. Although these scenes corrected with the GLS control are filtered to have cloud and snow cover scores of less than 4%, as was done with the Geometric Supersite imagery, the GLS assessments lacks the more careful visual inspection of the imagery and final results generated from the imagery. Therefore, the GLS based results are more of an indication of both the spacecraft and instrument performance with the additional aspect of the less accurate set of ground control and error involving the mensuration step to that ground control. Due to the less stringent outlier rejection with the GLS ground control, and it not being as accurate as the Geometric Supersite control, when strictly considering the spacecraft and instrument performance the GLS derived results are not included in the assessment or as a measurement against the L9 system meeting its established set of geometric requirements. The GLS ground control can provide a verification of any gross errors present within the data, or as more of a sanity check of the overall results determined from the more accurate control.

For the results based on both sets of ground control, along with the screening of imagery for clouds scores and snow cover of less than 4%, each scene must keep at least 50 ground control points in the precision solution step to be considered within the assessments. This criterion of keeping at least 50 GCPs in the precision solution step helps ensure that GCPs are distributed throughout the imagery providing further confidence in the quality of the precision solution and geodetic accuracy results generated. Based on the Geometric Supersite control the absolute geodetic accuracy for OLI-2 was determined as 13.40 m CE90 and the relative geodetic accuracy was 5.42 m CE90. This assessment was determined from the carefully selected 62 scenes acquired over the Geometric Supersites. The requirements for absolute and relative geodetic accuracy for L9 are 65- and 25-m CE90 respectively. Based on the GLS ground control the absolute geodetic accuracy was determined as 17.48 m CE90 and the relative geodetic accuracy was 6.46 m CE90. This assessment was based on 4335 images acquired. The higher GLS geodetic accuracy numbers are due to the control being less accurate, less stringent scene outlier rejection and the inclusion of high-latitude scenes which are not part of the Geometric Supersites and will tend to drive up in magnitude of the results due to winter conditions such as snow and ice affecting the registration results. Although some outlier logic based on snow cover and latitude

was performed on the GLS related data, poor correlations due to winter type landscape conditions, regardless of the snow cover, still affect the geodetic accuracy results for the GLS control and as neither the cloud or snow cover scores are 100% accurate, a few cloudy and snowy scenes will show up in these results. The individual scene derived numbers for the absolute geodetic accuracy based on the Geometric Supersites are shown in Figure 17 along with the requirement or specification. For comparison purposes the absolute geodetic accuracy results derived from the GLS control are shown in Figure 18. The GLS related numbers are an indication of what can be expected for the absolute geodetic accuracy of the L9 products.

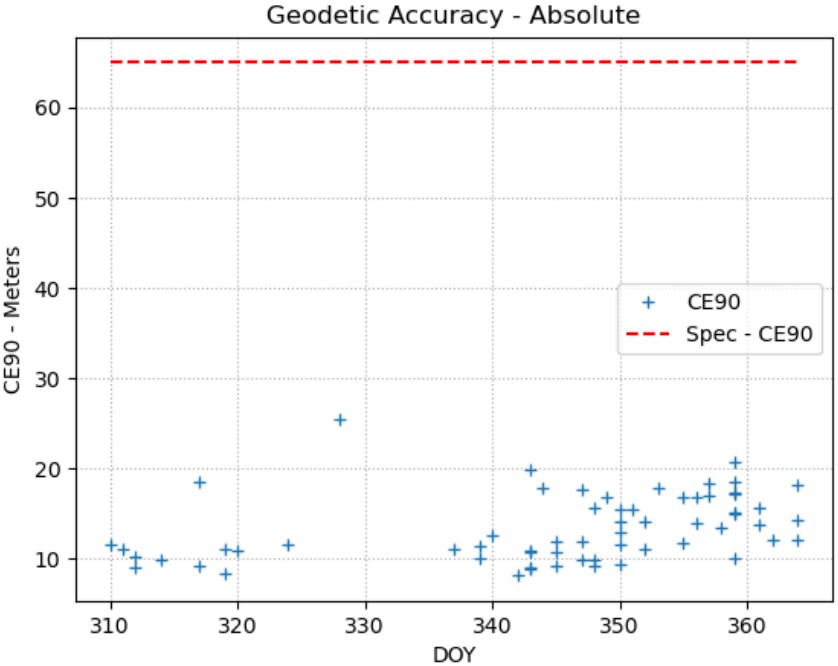

**Figure 17.** Absolute geodetic accuracy results for the Operational Land Imager-2 based on the Geometric Supersite control. Scenes were acquired in late 2021 during the spacecraft and instrument commissioning time frame. The day of year (DOY) during that commissioning period is shown on the *x*-axis. The measured Circular Error 90% (CE90) is shown along with the instrument requirement or specification (Spec) of 65 m which is also defined in terms of CE90 and plotted as a red dotted line within the figure.

The precision solution pre-fit standard deviation results, which determines the relative accuracy of the instruments, for the along and across track direction are shown for the Geometric Supersites in Figure 19. For comparisons purposes the pre-fit standard deviations using the GLS control are show in Figure 20. The results based on the GLS control are an indication of what can be expected for the relative geodetic accuracy of the L9 products, where the ground control plays a larger role in the results. The larger error present in the GLS data are more of indication of the mensuration issues with imagery, such as cloud or snow cover affecting the ability to measure differences between ground control and imagery, rather than an indication of the instrument or spacecraft performance. The noticeable gap in geodetic accuracy results in the plot for Figure 20 was due to an issue with the Solid-State Recorder onboard L9 and was associated with the reading and writing to bad memory blocks. Imaging was suspended during a brief period while investigations ensued. After thorough analysis, operational and software changes aboard the spacecraft allowed for mitigations to the impact of these bad memory blocks when performing image collects.

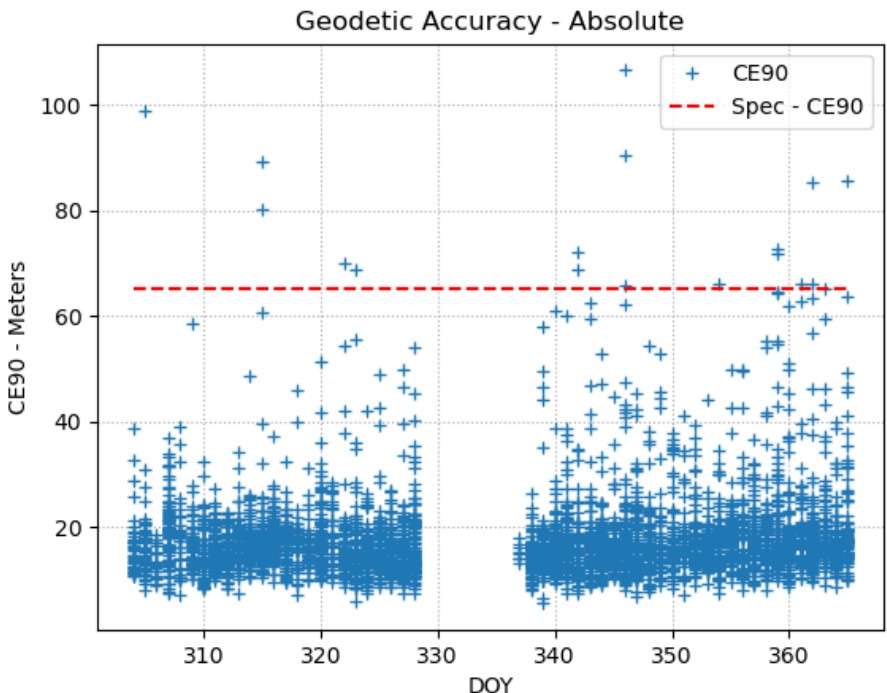

**Figure 18.** Absolute geodetic accuracy results for the Operational Land Imager-2 based on the Landsat Global Land Survey control. Scenes were acquired in late 2021 during the spacecraft and instrument commissioning time frame. The day of year (DOY) during that commissioning period is shown on the *x*-axis. The measured Circular Error 90% (CE90) is shown along with the instrument requirement or specification (Spec) of 65 m which is also defined in terms of CE90 and plotted as a red dotted line within the figure.

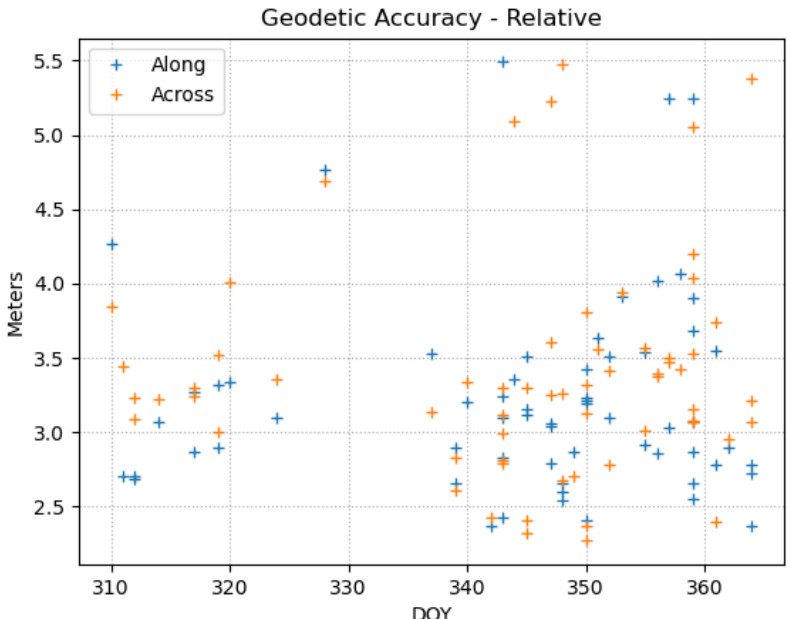

**Figure 19.** Relative geodetic accuracy pre-fit standard deviation results for the Operational Land Imager-2 based on the Geometric Supersite control. Scenes were acquired in late 2021 during the spacecraft and instrument commissioning time frame. The day of year (DOY) during that commissioning period is shown on the *x*-axis. The pre-fit standard deviation numbers are used to determine the relative, within scene, accuracy of the instrument and data products.

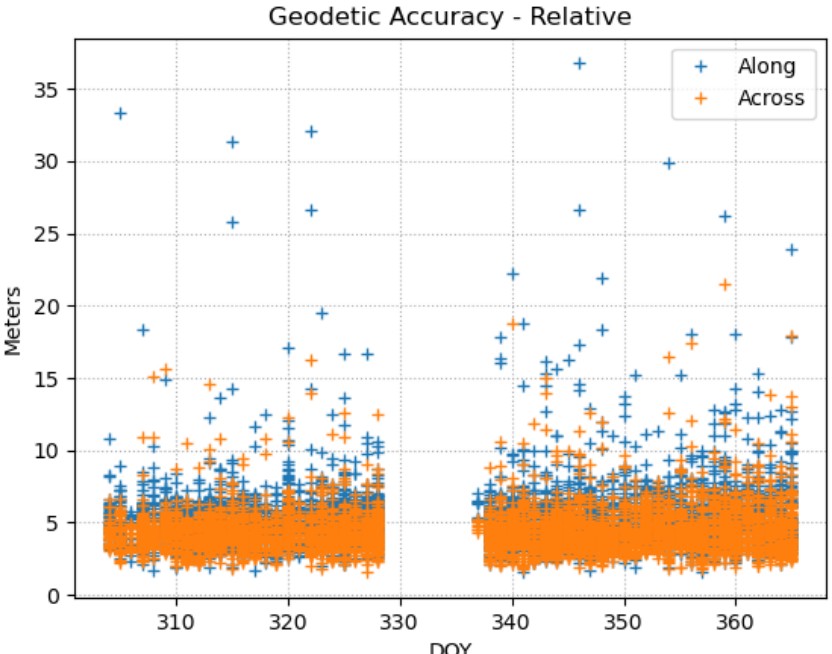

**Figure 20.** Relative geodetic accuracy pre-fit standard deviation results for the Operational Land Imager-2 based on the Global Land Survey ground control. Scenes were acquired in late 2021 during the spacecraft and instrument commissioning time frame. The day of year (DOY) during that commissioning period is shown on the *x*-axis.

The OLI-2 Geodetic Accuracy results for the differing sets of control are listed in Table 2. The difference, between the geodetic accuracy using both sets of control, allows for a comparison of the difference in the accuracy between the Geometric Supersite and GLS control. This difference, as shown in the last row of the table, is just over 4 m.

**Table 2.** Geodetic Accuracy for the Operational Land Imager-2 aboard the Landsat 9 spacecraft. Results based on only the Global Land Survey (GLS) Ground Control and those using only Geometric Supersites control are shown in the table. The last row in the table, labelled as Difference shows the difference in geodetic accuracy results between the Geometric Supersite and GLS control.

| Geodetic Accuracy (m) | | |
|---|---|---|
| **Type of Control** | **Absolute** | **Relative** |
| GLS (All) | 17.481 | 6.467 |
| Geometric Supersites | 13.406 | 5.429 |
| Difference | 4.075 | 1.038 |

For the geodetic accuracy assessment of TIRS-2, as there is not a set of thermal ground control points within the IAS or the Landsat Product Ground System (LPGS), LPGS being the USGS product generation system, that can be used for a direct comparison to the instrument its geodetic accuracy is determined analytically. This analytical approach is performed by Root Sum Squaring (RSS) of the OLI-2 geodetic accuracy assessment, OLI-2 band registration accuracy, the TIRS-2 to OLI-2 registration accuracy and the TIRS-2 band registration accuracy. Using the OLI-2 geodetic accuracy results based on the Geometric Supersites; (1) the OLI-2 band registration accuracy, (2) the TIRS-2 to OLI-2 registration accuracy, (3) the TIRS-2 band registration accuracy, the TIRS-2 Geodetic Accuracy results are calculated to be 26.88 m CE90. This flow of the numbers, which is the RSS'ing of the individual components, used in determining the TIRS-2 geodetic accuracy results shown in

Table 3. The band alignment results, listed in the table, are discussed in the band-to-band section that follows.

**Table 3.** Thermal Infrared Sensor-2 (TIRS-2) geodetic accuracy results. The USGS Image Assessment System (IAS) does contain any thermal ground control chips, therefore the TIRS-2 geodetic accuracy results are determined by analysis through other measurements made within the IAS. Note that the band registration accuracies in the table have been converted from Linear Error 90% LE90 to Circular Error 90% CE90.

| Contribution | Value | Units | Type | Value | Units | Type |
|:---:|:---:|:---:|:---:|:---:|:---:|:---:|
| OLI-2 geodetic accuracy | 13.41 | m | CE90 | 13.41 | m | CE90 |
| OLI-2 band registration accuracy | 3.18 | m | LE90 | 4.15 | m | CE90 |
| TIRS-2 to OLI-2 registration accuracy | 16.23 | m | LE90 | 21.18 | m | CE90 |
| TIRS-2 band registration accuracy | 6.72 | m | LE90 | 8.77 | m | CE90 |
| TIRS-2 geodetic accuracy performance | | | | 26.88 | m | CE90 |
| TIRS-2 geodetic accuracy requirement | | | | 76.00 | m | CE90 |

The band-to-band registration accuracy specification defines the accuracy with which corresponding Level 1TP pixels, for both the OLI-2 and TIRS-2 instruments, have their spectral bands co-aligned. It is important to note that the band-to-band requirement applies to Level 1TP images, after the precision-terrain correction has been applied to the data, including image resampling. Neither sensor's architecture provides inherent registration between spectral bands due to the along track displacement of the individual bands and SCAs, without precision and terrain corrections being applied. For band-to-band analysis 423 non-cirrus image data sets where used, while 49 image data sets were used in the analysis involving the cirrus band. The Landsat cirrus band (1.360–1390 μm) detects high-altitude cloud contamination that may not be visible in other bands. A band-to-band assessment of the OLI-2 instrument can be performed that includes the addition of the cirrus band, by choosing Landsat scenes with high altitude, cloud-free cirrus band scene content. However, this type of adequate cirrus band scenes is limited, which leads to the lowering the number of scenes for which a band-to-band assessment can be performed when this band is included in the analysis [25]. When the cirrus band is not used in the analysis of the OLI-2 band registration accuracy, 3.18 m in the line direction and 2.99 m in sample direction LE90 were determined. When the cirrus band is used in the analysis 3.42 m in the line direction and 3.10 m in the sample direction LE90 were determined. For TIRS-2, where only the two bands are present, 6.724 m in the line direction and 6.44 m in the sample direction LE90 was determined. For the TIRS-2 to OLI-2 alignment assessment, the same procedures that are used to determine the co-registration of the bands within instruments is performed, only in this case the band-to-band assessment is between the two instruments rather than within a given instrument. For this assessment the TIRS-2 imagery is resampled to the native 30-m pixel size of the OLI-2 instrument. It is worth noting that this upsampling of the TIRS-2 imagery will produce an additional amount of uncertainty to these results when comparing their result to the OLI-2 band-to-band results due to the lower resolution of the TIRS-2 instrument. The TIRS-2 to OLI-2 registration accuracy assessment, based on 427 scenes, was determined to be 16.23 m in the line direction and 15.92 m in the sample direction LE90. Figure 21 shows the OLI-2 band registration assessment for each band combination along with the band registration accuracy requirement of 4.5 m LE90 which is shown as a red dotted line. Figure 22 shows both the within band registration accuracy assessment of the two TIRS-2 bands and the TIRS-2 to OLI-2, minus the cirrus

band, band registration assessment. The TIRS-2 to OLI-2 registration requirement of 30 m LE90 is shown as a red dotted line. The TIRS-2 within band registration requirement is 18 m LE90. The within instrument and between instrument band registration is achieved for all bands for L9 (Figures 21 and 22).

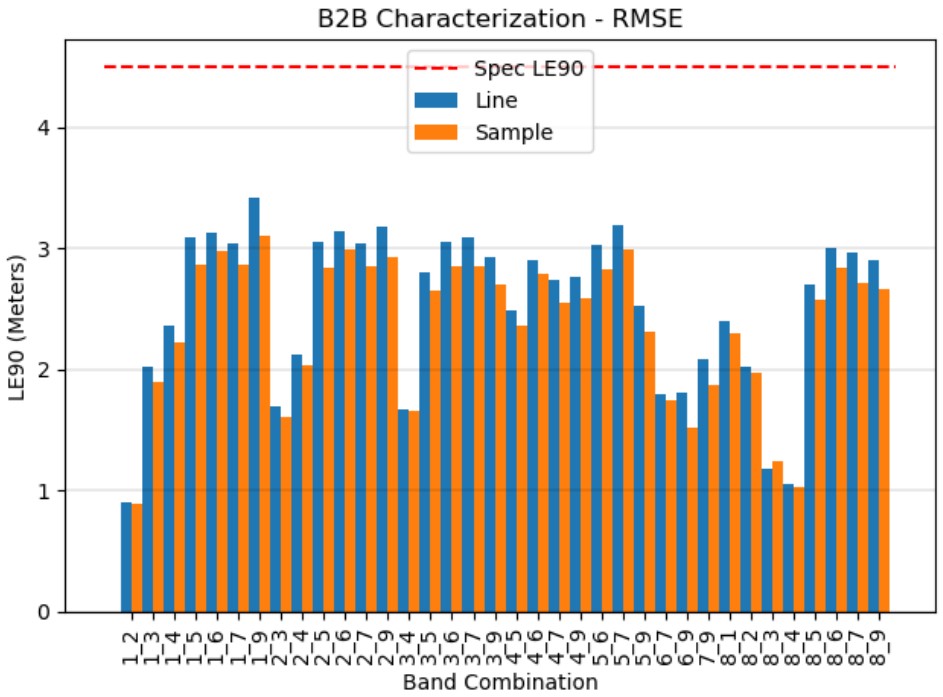

**Figure 21.** Operational Land Image-2 band registration accuracy assessment. Each band combination is assessed. The band-to-band requirement or specification (Spec), absent an assessment involving the cirrus band, is 4.5 m Linear Error 90% (LE90) which is shown as a dotted red line in the plot.

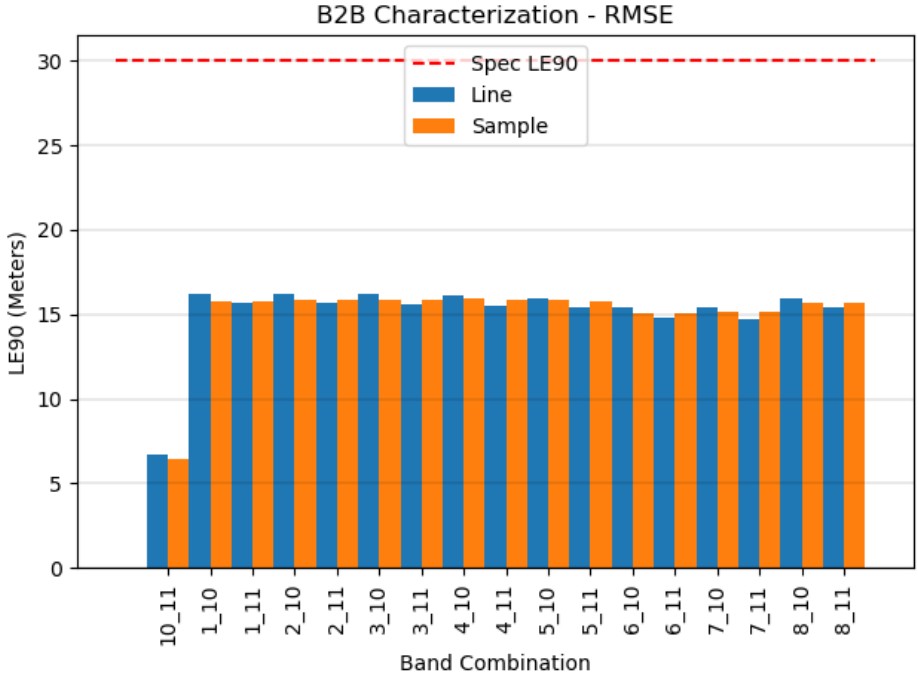

**Figure 22.** Thermal Infrared Sensor-2 (TIRS-2) within band registration accuracy assessment along with TIRS-2 to Operational Land Imager-2 (OLI-2) between instrument, per band, registration accuracy assessment. The TIRS-2 to OLI-2 band registration requirement or specification (Spec) which is 30-m Linear Error 90% (LE90) is shown as a dotted red line in the plot.

The Geometric Accuracy Assessment is the registration accuracy of the L1TP imagery. Both the Geometric Supersites and GLS were used in separate geometric assessments. In each case a group of the ground control, either Geometric Supersite or GLS, is used in the registration step, while a separate set of control from the same ground control type is used with the L1TP to determine how well the precision-terrain product is registered to the same type of control used for registration. Based on the Geometric Supersite control the geometric accuracy for OLI-2 was determined as 3.72 m CE90. This assessment was determined from the carefully selected 36 scenes acquired over the Geometric Supersites. The requirements for geometric accuracy for L9 are 12-m CE90. Based on the GLS ground control the geometric accuracy was determined to be 8.12 m CE90. This assessment was based on 4020 images acquired. These two sets of geometric accuracy are shown in Figures 23 and 24.

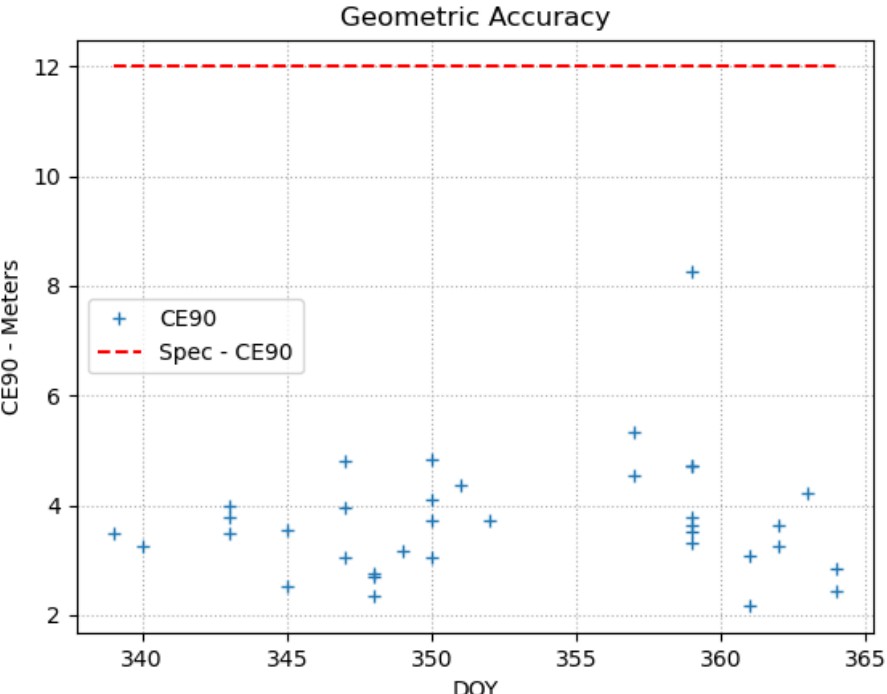

**Figure 23.** Operational Land Imager-2 geometry accuracy results based on Geometric Supersites. Scenes were acquired during the Landsat 9 commissioning period. Scenes with less than 4% cloud cover and that kept more than 50 ground control points are shown in the plot. The Landsat 9 requirement or specification (Spec) for geometric accuracy of 12 m Circular Error 90% (CE90) is shown in the plot as a dotted red line.

Table 4 shows the geometric accuracy for both the Geometric Supersites and GLS control. The Geometric Supersite scenes were chosen to have less than 4% cloud cover and contain little to no snow or ice. The GLS results shown uses imagery that has less than 4% cloud cover, less than 4% snow cover, are not acquisitions acquired at high latitudes, and whose post fit geodetic accuracy means and standard deviations are below the threshold of 30 m associated with an L1TP product definition [26]. Of note, the geometric accuracy results as less stringent with respect to the outlier logic applied to their results when compared to the geodetic accuracy results, thus their results often show higher residuals in locations with high cloud or snow cover [27] The geometric accuracy using the GLS only acquired over the Geometric Supersite WRS-2 path and rows is shown in the table along with the difference between these results and the Geometric Supersite results over the same WRS-2 path and rows. This difference shows there is small difference in the absolute accuracy of the two, with that difference being a little greater than 4 m.

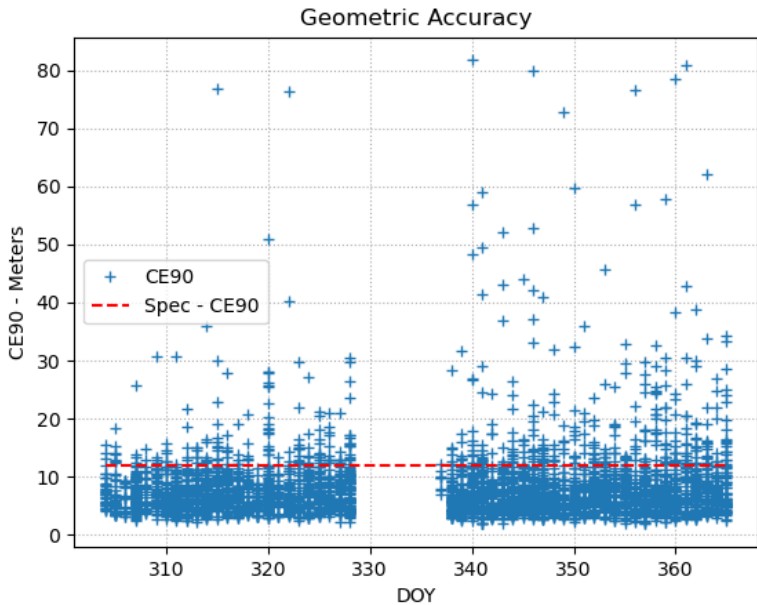

**Figure 24.** Operational Land Image-2 (OLI-2) geometric accuracy results based on Global Land Survey ground control. Scenes were acquired during the Landsat 9 commissioning period. Scenes with less than 4% cloud cover, less 4% snow cover, were acquired at mid-to-low latitudes, were consistent with the 30-m Level-1 Terrain Precision threshold and that kept at least 50 points in the mensuration step are shown in the plot. The Landsat 9 requirement of specification (Spec) for geometric accuracy of 12 m Circular Error 90% (CE90) is shown in the plot as a dotted red line.

**Table 4.** Operational Land Image-2 geometric accuracy. Results based on only the Global Land Survey (GLS) Ground Control and the Geometric Supersites are shown in the table. The last row in the table, labelled as Difference shows the difference in geometric accuracy results between the Geometric Supersite and GLS control.

| Type of Control | Geometric Accuracy CE90 (m) |
|---|---|
| GLS | 8.123 |
| Geometric Supersites | 3.727 |
| Difference | 4.395 |

Like the geodetic accuracy for TIRS-2, the geometric accuracy for TIRS-2 is derived by analysis of the other calibration components performed. This is due to the same reason as that of the geodetic accuracy, the IAS does not contain thermal ground control chips for a direct comparison between the thermal bands and a set of reference imagery. Using the OLI-2 geometric accuracy results based on the Geometric Supersites, the TIRS-2 geometric accuracy results are calculated as 23.59 m CE90. The flow of the numbers determining the TIRS-2 geometric accuracy results are shown in Table 5.

**Table 5.** Thermal Infrared Sensor-2 geometric accuracy results. The USGS Image Assessment System does not contain any thermal ground control chips, therefore the TIRS-2 geometric accuracy results are determined by analysis through other measurements made within the IAS, which are tied to the Geometric Supersite ground control.

| Contribution | Value | Units | Type | Value | Units | Type |
|---|---|---|---|---|---|---|
| OLI-2 geometric accuracy | 3.73 | m | CE90 | 3.73 | m | CE90 |
| OLI-2 band registration accuracy | 3.18 | m | LE90 | 4.15 | m | CE90 |

**Table 5.** *Cont.*

| Contribution | Value | Units | Type | Value | Units | Type |
|---|---|---|---|---|---|---|
| TIRS-2 to OLI-2 registration accuracy | 16.23 | m | LE90 | 21.18 | m | CE90 |
| TIRS-2 band registration accuracy | 6.72 | m | LE90 | 8.77 | m | CE90 |
| TIRS-2 L1T geometric accuracy performance | | | | 23.59 | m | CE90 |
| TIRS-2 L1T geometric accuracy requirement | | | | 42.00 | m | CE90 |

## 4. Conclusions

Landsat 9 was launched on 27 September 2021. The spacecraft's payloads included the Operational Land Imager-2 (OLI-2) and Thermal Infrared Sensor-2 (TIRS-2). The geometric commissioning period, the time frame that typically contains the largest geometric calibration parameter changes, went from several weeks after launch until the end of 2021. During this time that new geometric calibration parameters were being determined, the geometric performance of the spacecraft and its instruments were also being assessed. The L9 spacecraft and instruments were determined to be meeting all key geometric performance requirements. These key geometric parameters and their assessed performance based on the orbit measurements and assessments are summarized in Table 6. The margins listed within the table are determined based on the difference between the measurement and requirement value.

$$Margin = (Measured - Requirement) / Requirement * 100\% \qquad (3)$$

**Table 6.** Landsat 9 spacecraft, Operational Land Imager-2 (OLI-2) and Thermal Infrared Sensor (TIRS-2) key geometric performance requirements. All key geometric performance requirements are being met.

| Requirement | Measured Value | Requirement Value | Units | Margin |
|---|---|---|---|---|
| OLI-2 Swath | 189.96 | 185 | km | 2.7% |
| TIRS-2 Swath | 186.66 | 185 | km | 0.9% |
| OLI-2 Band Registration Accuracy (no cirrus) | 3.19 | 4.5 | m (LE90) | 29.2% |
| OLI-2 Absolute Geodetic Accuracy | 13.41 | 65 | m (CE90) | 79.4% |
| OLI-2 Relative Geodetic Accuracy | 5.43 | 25 | m (CE90) | 78.3% |
| OLI-2 Geometric (L1T) Accuracy | 3.73 | 12 | m (CE90) | 68.9% |
| TIRS-2 Band Registration Accuracy | 6.72 | 18 | m (LE90) | 62.6% |
| TIRS-2 to OLI-2 Registration Accuracy | 16.23 | 30 | m (LE90) | 45.9% |
| TIRS-2 Absolute Geodetic Accuracy | 26.88 | 76 | m (CE90) | 64.6% |
| TIRS-2 Geometric (L1T) Accuracy | 23.59 | 42 | m (CE90) | 43.8% |

**Author Contributions:** Methodology, M.J.C., R.R. and J.C.S.; Validation, M.L. All authors have read and agreed to the published version of the manuscript.

**Funding:** Work performed under USGS contract G15PC00012.

**Data Availability Statement:** The Level-1 products discussed and used within the analysis of this paper can be accessed through Earth Explorer https://earthexplorer.usgs.gov/ (accessed 4 April 2023).

**Conflicts of Interest:** The authors declare no conflict of interest.

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
