# Peer review of "Landsat 9 Geometric Commissioning Calibration Updates and System Performance Assessment"

_remotesensing, doi:10.3390/rs15143524_

Round 1

Reviewer 1 Report

I enjoyed reading this article and learning about the methodologies and subsequent on-orbit results of the Landsat 9 geometric calibration.  In general, the article is well-written aside from minor typos throughout, especially in the latter half of the article.  The suggestions below should help complete the manuscript for publication.

·      Figure 3:  Consider removing this table/figure and just talking to these numbers from within the text.  Note that (I think) the caption has typos in it as well.

·      Section 1.2 needs to be heavily referenced so an interested reader can better understand the nuances of the model and recreate the results.  

·      Line 270:  Please articulate (with a sentence or two) why the SWIR-2 band is chosen for TIRS-2 alignment when so much emphasis is placed on Pan-band Cal.

·      Redo Figures 6 & 9...they are too pixelated.  Figure 9 is a screenshot from envi.

·      Can you discuss why the mean-deltas for TIRS-2 (Figures 14 & 15) are, wrt pixel IFOV, so much larger than OLI-2?

·      Please articulate how are the margins in Table 6 calculated?

Author Response

Thanks for taking the time to review the paper.  Your comments were very helpful.  I attached a response to comments in the uploaded word document.

Reviewer 2 Report

This manuscript provides an update on the geometric calibration of Landsat 9 during the commissioning phase of the satellite. There are some minor issues that need to be addressed before publication: 

·      Figure 3 should be “Table 1.”

·      Line 176: Can you define or elaborate more on CE90 and LE90 (or provide citation for it).

·      Line 186: Can you provide more explanation on what makes an image a “Geometric Supersite.” Is it the 6-meter accuracy mentioned in line 216? Were these sites used for L8 commissioning as well?

·      Line 310: During commissioning, what was the orbit height (compared to the WRS-2 grid), did that any way impact the geometric calibration process?

·      Figure 17: Can you mention the std dev. For the TIRS-2 to OLI-2 alignment, and whether is it within the expected range.

Author Response

Thanks for taking the time to review the paper.  Your comments were very helpful.  Responses to comments were uploaded to a word document.

Reviewer 3 Report

Dear Authors,

The paper deals with the geometric characterizations, calibrations, and performance analyses conducted during the commissioning period of the Landsat 9 spacecraft and its instruments. This research is valuable for the scientists, who work on geometric calibrations of satellite data, and since Landsat 9 is the last mission of the Landsat program, extensive studies on its performance are welcome. The paper is well-prepared and well-organized. I just have one comment in minor importance that major quantitative results should be given in the abstract.

The contribution of the study to the current literature is worthy of note since it is based on the literature gaps.The conclusion is consistent with the evidence and arguments presented and they address the main question posed. Figures, tables and references are appropriate.

All the best.

Author Response

Thanks for taking the time to review the paper.  I responses to comments were uploaded to a word document.
